# Early Detection of Powdery Mildew Disease and Accurate Quantification of Its Severity Using Hyperspectral Images in Wheat

**Imran Haider Khan [1,†], Haiyan Liu [1,†], Wei Li [1], Aizhong Cao [2], Xue Wang [1], Hongyan Liu [1], Tao Cheng [1], Yongchao Tian [1], Yan Zhu [1], Weixing Cao [1] and Xia Yao [1,\*]**

1   National Engineering and Technology Center for Information Agriculture, Key Laboratory for Crop System Analysis and Decision Making, Ministry of Agriculture, Jiangsu Key Laboratory for Information Agriculture, Jiangsu Collaborative Innovation Center for Modern Crop Production, Nanjing Agricultural University, Nanjing 210095, China; 2018101174@njau.edu.cn (I.H.K.); 2018101173@njau.edu.cn (H.L.); 2020201096@stu.njau.edu.cn (W.L.); wangxue@njau.edu.cn (X.W.); 2016101049@njau.edu.cn (H.L.); tcheng@njau.edu.cn (T.C.); yctian@njau.edu.cn (Y.T.); yanzhu@njau.edu.cn (Y.Z.); caow@njau.edu.cn (W.C.)
2   National Key Laboratory of Crop Genetics and Germplasm Enhancement, Cytogenetics Institute, Nanjing Agricultural University/JCIC-MCP, Nanjing 210095, China; caoaz@njau.edu.cn
\*   Correspondence: yaoxia@njau.edu.cn; Tel.: +86-25-84396565; Fax: +86-25-84396672
†   Imran Haider Khan and Haiyan Liu are the co-first author.

**Abstract:** Early detection of the crop disease using agricultural remote sensing is crucial as a precaution against its spread. However, the traditional method, relying on the disease symptoms, is lagging. Here, an early detection model using machine learning with hyperspectral images is presented. This study first extracted the normalized difference texture indices (NDTIs) and vegetation indices (VIs) to enhance the difference between healthy and powdery mildew wheat. Then, a partial least-squares linear discrimination analysis was applied to detect powdery mildew with the combined optimal features (i.e., VIs & NDTIs). Further, a regression model on the partial least-squares regression was developed to estimate disease severity (DS). The results show that the discriminant model with the combined VIs & NDTIs improved the ability for early identification of the infected leaves, with an overall accuracy value and Kappa coefficient over 82.35% and 0.56 respectively, and with inconspicuous symptoms which were difficult to identify as symptoms of the disease using the traditional method. Furthermore, the calibrated and validated DS estimation model reached good performance as the coefficient of determination ($R^2$) was over 0.748 and 0.722, respectively. Therefore, this methodology for detection, as well as the quantification model, is promising for early disease detection in crops.

**Keywords:** wheat powdery mildew; hyperspectral imaging; early; detect the crop disease; quantify the disease severity

## 1. Introduction

Wheat (Triticum aestivum) is an important food crop in China. The sustainable production of wheat is important for the stability of China's society and economy. However, wheat is often attacked by various diseases due to unfavorable environments and management conditions. Powdery mildew (PM) (*Blumeria graminis f.* sp. tritici) is one of the major diseases that seriously affect wheat yield and quality [1]. Currently, people use disease-resistant varieties [2,3] and spray pesticides [4] to control PM, but these are not optimal solutions, and will produce an increased cost and environmental pollution. The essential approach to combating PM is the early identification and quantitative assessment of disease severity (DS), thereby helping farmers to ensure timely use of fungicides [5].

Traditional methods to identify the disease mainly depend on visual inspection and laboratory tests. Farmers usually visually judge its presence in the field, but there are

frequent mistakes. Laboratory tests include enzyme-linked immunosorbent assay (ELISA), immunofluorescence, fluorescence in situ hybridization (FISH), and polymerase chain reaction (PCR) methods, which are more accurate than visual inspection but usually lag and require precision instruments and strict operations, therefore, most are time consuming.

With the development of spectroscopy technologies, spectral identification methods (Mutka and Bart, 2015) have achieved remarkable results in the field of intelligent identification of plant diseases, such as RGB imagery and multi/hyperspectral technology [6]. A disease identification method using RGB imagery is easier to implement and does not require sophisticated chemical analysis instruments or expensive sensors, imaging equipment, etc. RGB imagery recognition manually extracts the recognizable features of each disease, such as color, shape, and texture features. These features can further be used to obtain a model for distinguishing disease types (Siricharoen) [7]. However, some factors seriously affect the application of visible imaging to plant disease including light condition, removal of the background, and other automatic image processing techniques (Barbedo) [8]. Multispectral images contain more band information compared with RGB images. Satellite imagery and images obtained from the combined use of a UAV with multispectral camera are two common ways to obtain multispectral image. This method seems to be one of the best methods for quickly and accurately detecting and mapping disease incidence over large areas [9]. However, it appears to be insufficient to utilize this combination of narrow bands for disease mapping at earlier stages of the disease's development [10].

Compared with the above methods, hyperspectral technology has a high cost and its complex data interpretation remains challenging [11]. However, its applications are mature and reliable, and have been widely used to monitor the physiological status of vegetation [12], drought stress [13], nutrient deficiencies [14], and disease stress [15]. The principle is that the interaction between plants and the electromagnetic spectrum relies on photosynthetic chlorophyll in the visible light band (VIS), leaf anatomy in the near-infrared band (NIR), and leaf water in the short-wave infrared band [16]. When wheat is subjected to PM stress, its spectral reflectance changes according to physiological and biochemical changes in its leaves, such as decreased chlorophyll content or destroyed cell structure [17]. When combined with aerial platforms, hyperspectral technology is very well-suited for field phenotyping due to its capability for the characterization of subtle details of diseases. Many studies monitoring plant diseases have been conducted using ground and aerial hyperspectral reflectance. For example, Shi et al. [18] proposed a spectral vegetation indices-based kernel discriminant approach for the detection and classification of yellow rust, aphid, and powdery mildew in winter wheat. Zheng et al. [19] developed optimal spectral indices with three-band combinations based on sensitive wavebands to detect yellow rust disease in wheat at different growth stages. However, by using only spectral information, those studies identified disease at the middle or late infection stages, thereby missing the critical time for prevention and control of the disease.

Hyperspectral imaging (HI) is a new nondestructive monitoring technology developed in recent years. It combines imaging with spectroscopic techniques to obtain both spectral and spatial information [20]. As well as the spectral information of HI, its spatial information offers an accurate and reliable resource to identify disease. Texture is a typical spatial information that refers to the recurring local patterns and permutation rules of pixels. The texture features can reflect changes in the intensity of pixels to distinguish and identify objects [21]. Analyzing texture information from images can detect foliar surface changes due to fungal disease. For example, Yang et al. [22] used gray histogram and a gray-level co-occurrence matrix to extract the texture features of rice blast leaves, establishing a stepwise discriminant model with a classification accuracy of more than 90%. Al-Saddik et al. [23] combined spectral and texture data to detect yellowness and esca in grapevines with an overall accuracy of 99% for both of the diseases. Although the above studies achieved some useful results, there is scant evidence regarding the detection of PM through the fusion of spectral and texture information, especially at the early stages of disease infection.

The main element of HI processing for the detection of disease at its early stages is feature engineering, which transforms the raw hyperspectral data into features suitable for modeling. Feature selection is one of the important methods of feature engineering for dimensionality reduction of hyperspectral data. Feature selection methods, such as stepwise discriminant analysis, select a subset of characteristic wavelengths that preserve certain information of the full spectrum. The selected wavelengths can be used to construct vegetation indices [24] and disease indices [25]. However, most of the feature-selection methods are based on a single dataset or a single model, which do not account for the impaction of the changes of sample or variable on variable selection. To solve these problems, introduced herein is sub-window permutation analysis (SPA) [26]. The main idea of SPA is to build hundreds of sub-models based on different sub-datasets by a random sampling of the total samples and variables, and then to test those models statistically to determine the importance of each variable. SPA is a simple and effective feature-selection method, which considers both the influence of the samples and interaction among the variables. The SPA method was first proposed for chemistry and life science analysis and has proved its advantage. Mo et al. [27] combined near infrared spectroscopy with SPA to select the important variables related to haloxyfop-p-methyl residue in edible oil from the whole band and established a PLS-LDA model with an accuracy of 94.74%. Sun et al. [28] used near infrared (NIR) spectroscopy and SPA to detect complex adulteration of camellia oil. In the field of disease detection, SPA was initiated in wheat PM and successfully extracted information from non-imaging hyperspectral data by Wang [29]. However, the point-by-point spectral characteristics of the entire wheat leaf cannot be analyzed due to the measurement characteristics of the analytical spectral device. In the early stage of wheat infection by powdery mildew, the disease spots are scattered and usually not evenly distributed. For this reason, the combination of spectral and imaging technology can provide a new perspective for the early diagnosis of wheat powdery mildew.

Here, the main objectives of this study were (I) to determine the sensitive features by SPA and combine spectral and texture analysis, (II) to develop a robust discriminant model to early detect wheat PM by PLS-LDA, and (III) to establish an accurate model to quantify disease severity of wheat PM by PLSR. This methodology is expected to provide a new technique to detect the disease. This study demonstrates the capability of hyperspectral imaging to determine occurrence and severity of PM infection at the early stage.

## 2. Materials and Methods

### 2.1. Experimental Site and Design

The experiment was carried out at the Pailou experiment station, Nanjing Agricultural University (32°1′N, 118°15′E) during the wheat growing season of 2017–2018. Nannong 0686 wheat with high susceptibility to disease was grown in greenhouses. We planted 24 pots (280 mm × 280 mm × 500 mm) with a density of ten seeds per pot. Eight pots were health for control, eight for inoculation at the jointing stage, and the residual eight pots were put separately in the greenhouse for inoculation at booting stage. At the two growth stages, the infected wheat strains were used to shake off spores to inoculate the wheat plant in the pot, then they were placed horizontally in the pot, and all of them were incubated in a controlled environment at a temperature of 25 °C and at 80% relative humidity for 72 h.

### 2.2. Definition of Disease Severity (DS)

The DS of wheat PM is defined as the percentage of the disease pustules portion relative to the total leaf area, which was recorded by visual assessment according to a protocol by Graeff et al. [30]. Because this method is subjective, the assessment was carried out by just one investigator to eliminate any bias due to individual differences. Apart from using DS to quantify the incidence of PM, the leaves are also classified into the categories of healthy leaves (DS less than 1%) and diseased leaves (DS more than 1%) for detection of wheat PM (Figure 1).

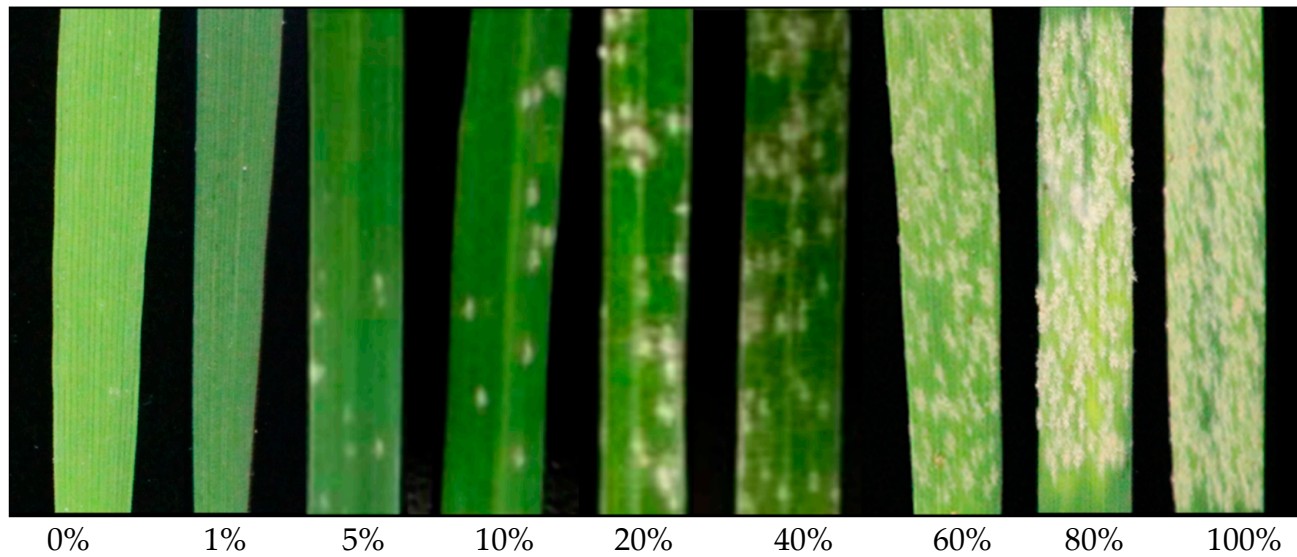

| 0% | 1% | 5% | 10% | 20% | 40% | 60% | 80% | 100% |

**Figure 1.** Images of wheat leaves with different DS.

### 2.3. Acquisition and Pre-Processing of Hyperspectral Images

In this study, we used the GaiaField portable HI system, consisting of an imaging lens coupled with an imaging spectrometer (V10E; Specim, Oulu, Finland) and a CCD camera (C8484-05; Hamamatsu Photonics, Osaka, Japan) (Figure 2). The HI system also included two light sources of 150 W halogen lamps (Oriel Instruments, Stratford, CT, USA) angled at 45°, a computer, a dark box, and a tripod. The spectral data were recorded in the VIS-NIR range of 400–1000 nm with a spectral resolution of 2.8 nm. The specification of each hyperspectral image was $1392 \times 1040$ (spatial dimensions) $\times$ 256 (spectral bands). The distance between the leaves and the camera was set as 300 mm and the exposure time was set as 0.13 s to obtain clear and non-deformable images. The data were processed and analyzed using the ENVI 5.2 (Exelis Visual Information Solutions, Boulder, CO, USA) and MATLAB R2014a (The MathWorks Inc., Natick, MA, USA) software packages.

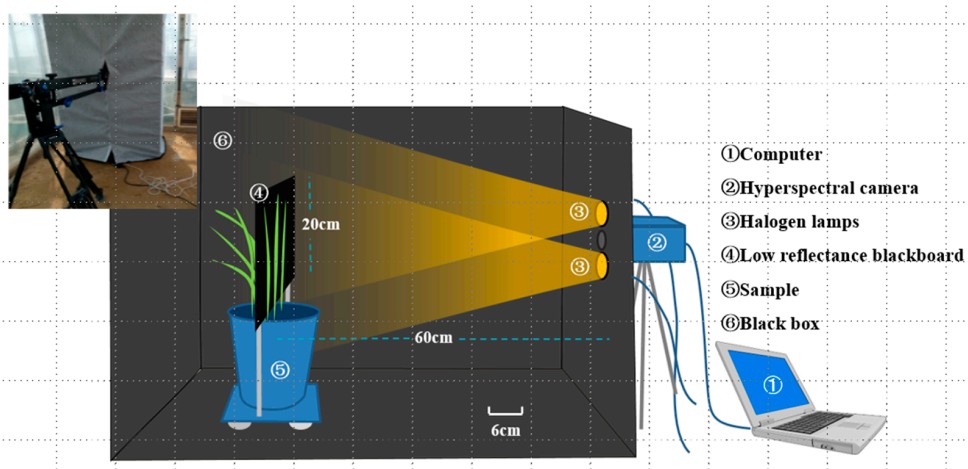

**Figure 2.** Test scenario of the hyperspectral imaging system.

The HI preprocessing steps included reflectance conversion, noise removal, and reflectance extraction. Because of the dark current and physical configuration of the imaging

system, some bands with weaker light intensity contain excessive noise. Therefore, a raw hyperspectral image was calibrated to reduce the noise according to the following equation:

$$I_c = \frac{I_\Gamma - I_d}{I_w - I_d} \tag{1}$$

where $I_c$ is the calibrated reflectance image, $I_r$ is the raw hyperspectral image, $I_d$ is the dark current image, and $I_w$ is the white reference image; $I_d$ is obtained by covering the camera lens with an opaque cap and turning off the light source, and $I_w$ is acquired by imaging a polytetrafluoroethylene white panel with spectral reflectance of 99%.

After obtaining the calibrated spectral reflectance, to smooth and minimize the noise signals in the images, the noise of the hyperspectral image data was reduced by using the minimum noise fraction [31]. The average hyperspectral reflectance of the whole leaf was directly extracted from an image of a single leaf.

### 2.4. Selection of the Sensitive Feature

In this study, an SPA algorithm was applied to determine the sensitive wavelength, and then texture features of the corresponding sensitive wavelength were calculated for developing the normalized difference texture index (NDTI).

### 2.4.1. Construction of Texture Indices

Hyperspectral image has a gray image in each of its spectral bands, so there is a large amount of redundant information. Based on the gray-level co-occurrence matrix (GLCM) of the gray image in each selected spectral band, eight texture features were acquired for each gray image [32], as listed in Table 1. In addition to the texture features, a new texture-based index called the normalized difference texture index (NDTI) was proposed, which follows the conventional definition of normalized difference vegetation index. A new NDTI was constructed with all possible two-texture feature combinations at all sensitive wavelengths with eight GLCM-based texture features of each gray image. The texture index NDTI has been used to classify sun and shade leaves of crops and plant biomass estimation [33]. For example, six wavelengths were selected in this research with 8 texture features at each wavelength, thus 48 texture features were acquired. NDTI can be calculated with the combination of two different texture features from all 48 texture features. A total of 2256 combination pairs were obtained, and the specific formula used for NDTI calculation was NDTI = (T1 − T2)/(T1 + T2), where T1 and T2 are the texture features of the selected sensitive wavelength.

**Table 1.** Texture features based on gray-level co-occurrence matrix.

| Number | Name | Equation | Description |
|:---:|:---:|:---:|:---:|
| 1 | Mean, MEA | $Mean = \sum\limits_{i,j=1}^{G} iP(i,j)$ | Reflects the average of grayscale |
| 2 | Variance, VAR | $Variance = \sum\limits_{i=1}^{G}\sum\limits_{j=1}^{G} (i-u)^2 P(i,j)$ | Reflects the size of the grayscale change |
| 3 | Homogeneity, HOM | $Homogeneity = \sum\limits_{i=1}^{G}\sum\limits_{j=1}^{G} \frac{P(i,j)}{1+(i-j)^2}$ | Reflects local homogeneity of texture |
| 4 | Contrast, CON | $Contrast = \sum\limits_{i=1}^{G}\sum\limits_{j=1}^{G} (i-j)^2 P(i,j)$ | Reflects the clarity of the texture |
| 5 | Dissimilarity, DIS | $Dissimilaity = \sum\limits_{i=1}^{G}\sum\limits_{j=1}^{G} P(i,j)|i-j|$ | Same as contrast, used to detect similarity |
| 6 | Entropy, ENT | $Entropy = -\sum\limits_{i=1}^{G}\sum\limits_{j=1}^{G} P(i,j)\log P(i,j)$ | Measures the amount of information of an image |
| 7 | Second Moment, SEM | $Second\ Moment = \sum\limits_{i=1}^{G}\sum\limits_{j=1}^{G} P^2(i,j)$ | Reflects the uniformity of the grayscale distribution of the image |
| 8 | Correlation, COR | $Correlation = \sum\limits_{i=1}^{G}\sum\limits_{j=1}^{G} \frac{(i-Mean_i)(j-Mean_j)P(i,j)}{\sqrt{Variance_i}\sqrt{Variance_j}}$ | Reflects the extension of a gray value along a certain direction |

Notes: $P(i,j) = V(i,j)/\sum_{i=0}^{G}\sum_{j=0}^{G} V(i,j)$ where $V(i,j)$ is the value in the cell at row of $i$ and column of $j$ and G is the number of rows or columns.

Here, the sensitive wavelengths for each texture feature were selected by SPA. SPA is a new statistics-based feature selection algorithm that combines a random-forest permutation method and model-based cluster analysis. The main aspects of SPA are the establishment of sub-models based on samples and variable sub-windows, and the subjection of parameters of interest to strict statistical analysis. The detailed implementation of SPA can be seen in Li et al. [26].

### 2.4.2. Selection of Vegetation Indices

When wheat is infected with PM the physiological and biochemical parameters of its leaves change accordingly [16], thereby changing its spectral reflectance. In total, 15 vegetation indices (VIs) frequently investigated in disease research were selected for this study, as listed in Table 2. And these 15 VIs are all highly related to the physiological and biochemical parameters.

**Table 2.** Vegetation indices used in this study.

| | Definition | Equations | Reference |
|---|---|---|---|
| 1. | Powdery mildew index, PMI | $(R515 - R698)/(R515 + R698) - 0.5 * R738$ | [34] |
| 2. | Modified simple ratio, MSR | $(R800/R670 - 1)/(R800/R670 + 1)^{1/2}$ | [35] |
| 3. | Photochemical reflectance index, PRI | $(R570 - R531)/(R570 + R531)$ | [36] |
| 4. | Photosynthetic radiation, PhRI | $(R550 - R531)/(R550 + R531)$ | [36] |
| 5. | Modified chlorophyll absorption ratio index, MCARI | $[(R701 - R671) - 0.2(R701 - R549)]/(R701/R671)$ | [37] |
| 6. | Anthocyanin reflectance index, ARI | $(R550)^{-1} - (R700)^{-1}$ | [38] |
| 7. | Structure independent pigment index, SIPI | $(R800 - R445)/(R800 - R680)$ | [39] |
| 8. | Normalized pigment chlorophyll ration index, NPCI | $(R680 - R430)/(R680 + R430)$ | [39] |
| 9. | Red-edge vegetation stress index, RVSI | $[(R712 + R752)/2] - R732$ | [40] |
| 10. | Narrow-band normalized difference vegetation index, NBNDVI | $(R850 - R680)/(R850 + R680)$ | [41] |
| 11. | Nitrogen reflectance index, NRI | $(R570 - R670)/(R570 + R670)$ | [42] |
| 12. | Triangular vegetation index, TVI | $0.5[120(R750 - R550) - 200(R670 - R550)]$ | [43] |
| 13. | Transformed chlorophyll absorption and reflectance index, TCARI | $3[(R700 - R670) - 0.2(R700 - R550)(R700/R670)]$ | [44] |
| 14. | Plant senescence reflectance index, PSRI | $(R680 - R500)/R750$ | [45] |
| 15. | Aphid index, AI | $(R740 - R887)/(R691 - R698)$ | [46] |

### 2.5. Development of the Recognition Model for Wheat Leaf Disease

The PLS-LDA method was used to classify the healthy and diseased wheat leaves with the sensitive features of vegetation and texture indices. PLS-LDA is effective in processing data such as small sample size, high dimensionality, and multicollinearity [39]. In the recognition model, the actual measured DS was quantitatively classified into two classes of healthy and diseased for discriminating PM infection. The model performance was evaluated using the overall accuracy (OA) derived from a confusion matrix and Kappa coefficient. An accurate model should have higher values of OA and Kappa coefficient.

### 2.6. Construction of the DS Estimation Model for Wheat Leaf Disease

Furthermore, partial least-squares regression (PLSR) was used to establish the regression relationships between the indices (VIs and NDTIs) and the DS for wheat leaves at different growth stages. As a mature multiple linear regression algorithm, PLSR is the simplest partial least-squares method, and has been widely used to analyze agricultural remote-sensing data [40]. In the severity model, the relationship between actual measured DS and indices was examined to evaluate the sensitivity of indices to PM at different growth stages. The performance of the wheat DS estimation model was evaluated by the coefficient of determination ($R^2$), the root mean square error (*RMSE*), and the relative root mean square error (*RRMSE*). $R^2$ can illustrate the correlation between VIs/NDTIs and DS. The robustness of models was assessed by both *RMSE* and *RRMSE*. The higher $R^2$ and the lower *RMSE* and *RRMSE* values, the better the model performs. $R^2$, *RMSE*, and *RRMSE* can be calculated as following equations:

$$R^2 = 1 - \frac{\sum_i \left(\hat{y}^i - y^i\right)^2}{\sum_i \left(\overline{y}^i - y^i\right)^2} \tag{2}$$

$$RMSE = \sqrt{\left(\frac{1}{M}\sum_1^M \left(y^i - \hat{y}^i\right)^2\right)} \tag{3}$$

$$RRMSE = \frac{RMSE}{\overline{y}^i} \tag{4}$$

where $y^i$ was the observed DS, $\hat{y}^i$ was the estimated DS, $\overline{y}^i$ was the mean of DS observation, and $M$ was the number of samples.

The calibration model was built using all data to achieve better performance, but the model may be overfitting. To avoid this phenomenon, 10-fold cross validation was adopted to assess the performance of models. In this process, the data were separated in 10 folds. The 9-fold data were used to build PLSR model. Then, the constructed model was used to predict the remaining 1-fold data. After 10 trials, all DS estimates can be obtained.

## 3. Results

### 3.1. Time-Series Variation of Spectral Reflectance

Figure 3 shows the spectral-reflectance changes of healthy and diseased leaves on different days after infection at the jointing stage. The average DS of the wheat leaves increased gradually over time. Compared with healthy leaves, the spectral reflectance of diseased leaves was higher in the VIS region but lower in the NIR region, which is consistent with previous studies [17]. Moreover, the spectral signature in the NIR region differed more obviously between normal and diseased leaves at late infection stage.

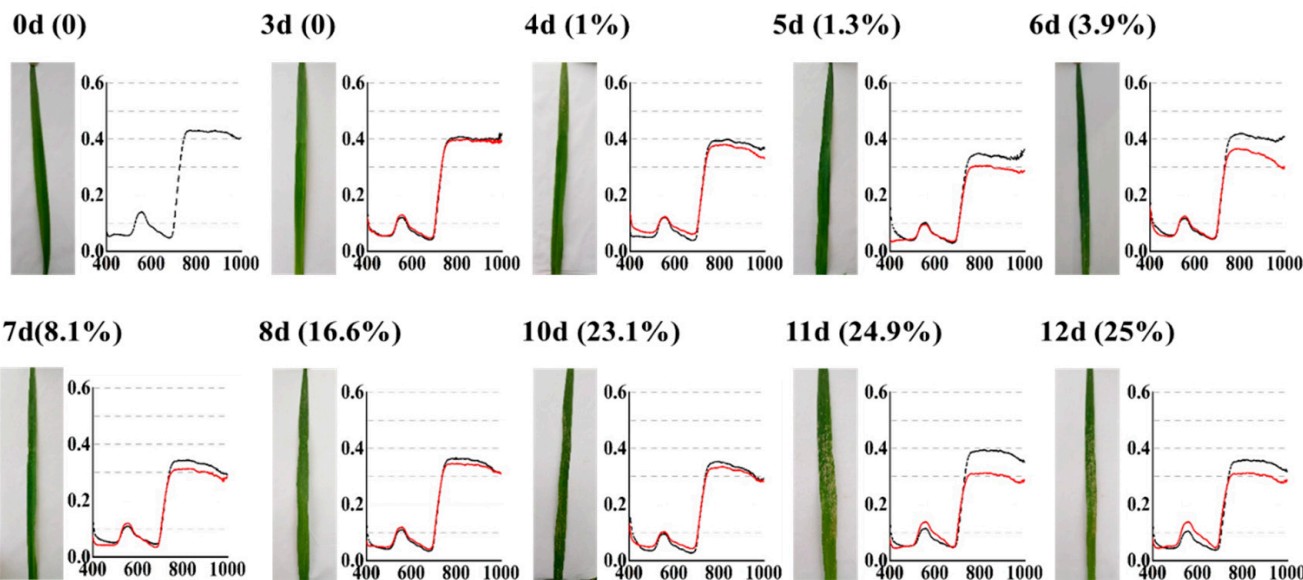

**Figure 3.** Reflectance changes of healthy (black line) and diseased (red line) wheat leaves after infection at the jointing stage. Note: X axis: Wavelength (nm); Y axis: reflectance; "12d (25%)" represents the 12th day after infection with an average DS of 25%.

### 3.2. Selection of the Sensitive Features

In this study, we used a new feature-selection algorithm SPA for selection of sensitive wavelengths for wheat PM, and then we calculated 8 texture features based on the selected sensitive wavelengths for developing the NDTIs. Meanwhile, sensitive VIs were selected and analyzed to compare with the performance of NDTIs. The results of feature selection are as the below.

### 3.2.1. Selection of Sensitive Wavebands

Due to the noisy spectral reflectance data around 400 and 1000 nm, the wavelengths used in this study were in the range of 437.2–976.2 nm. By using an SPA algorithm, we obtained the conditional synergetic score (COSS) values of each wavelength, indicating its sensitivity to PM, as shown in Figure 4A. The higher the COSS value, the more the waveband is sensitive to disease. Then we ranked the spectral bands in descending order according to their COSS values and selected the top 20 bands further for construction of PLS-LDA models. We successively added these bands to the PLS-LDA model one by one according to their COSS values in descending order and recorded their calibration and validation accuracies for selection of the best combination of sensitive wavebands. These calibration and leave-one-out cross validation (LOOCV) accuracies are shown in Figure 4B. Cross validation accuracies gradually increased and achieved highest coinciding point at the sixth waveband, hence, top six wavebands with higher COSS values were selected as sensitive wavelengths, which were 533.5, 659.2, 528.5, 661.7, 968.2, and 523.6 nm by descending COSS values.

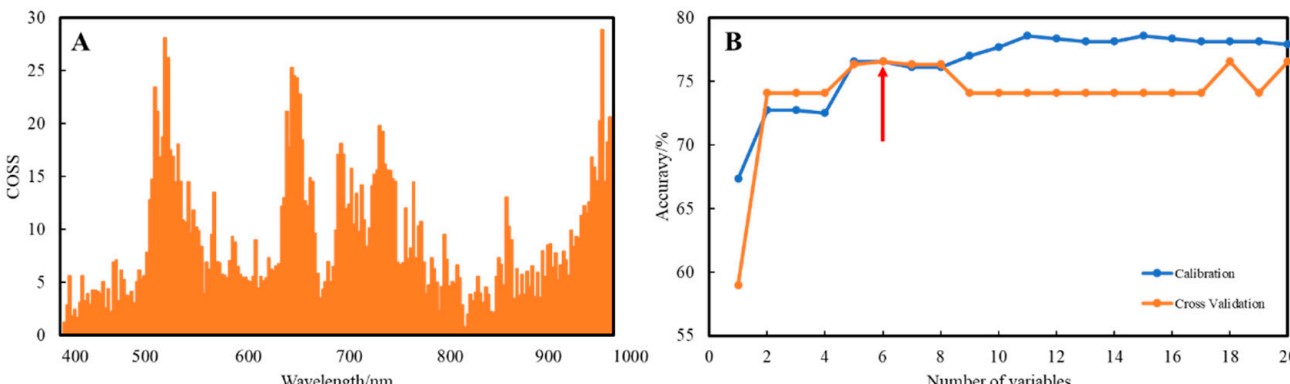

**Figure 4.** (**A**): COSS value computed by SPA for each waveband. (**B**): Calibration and cross validation classification accuracies at different number of variables based on PLS-LDA.

### 3.2.2. Selection of Optimal Vegetation Indices

The sensitive VIs were selected by correlation analysis between the VIs and DS of wheat PM in two growth stages. Figure 5 summarizes the sensitivity of 15 VIs to DS at each of the two individual growth stages and at both growth stages as a whole. Except for TVI, these selected VIs were significantly correlated to DS at both stages. The response of VIs to DS at different growth stages differed from that of both stages. Most of the VIs showed high correlation to DS at booting stage, while only a few showed good relationships with DS at the jointing stage. The response of the VIs in different growth stages is not the same. In total, six spectral vegetation indices (PMI, MSR, MCARI, ARI, SIPI, and PSRI) were significantly correlated with DS at each growth stage as well as for both stages as a whole. They were selected for further disease recognition regression and classification models.

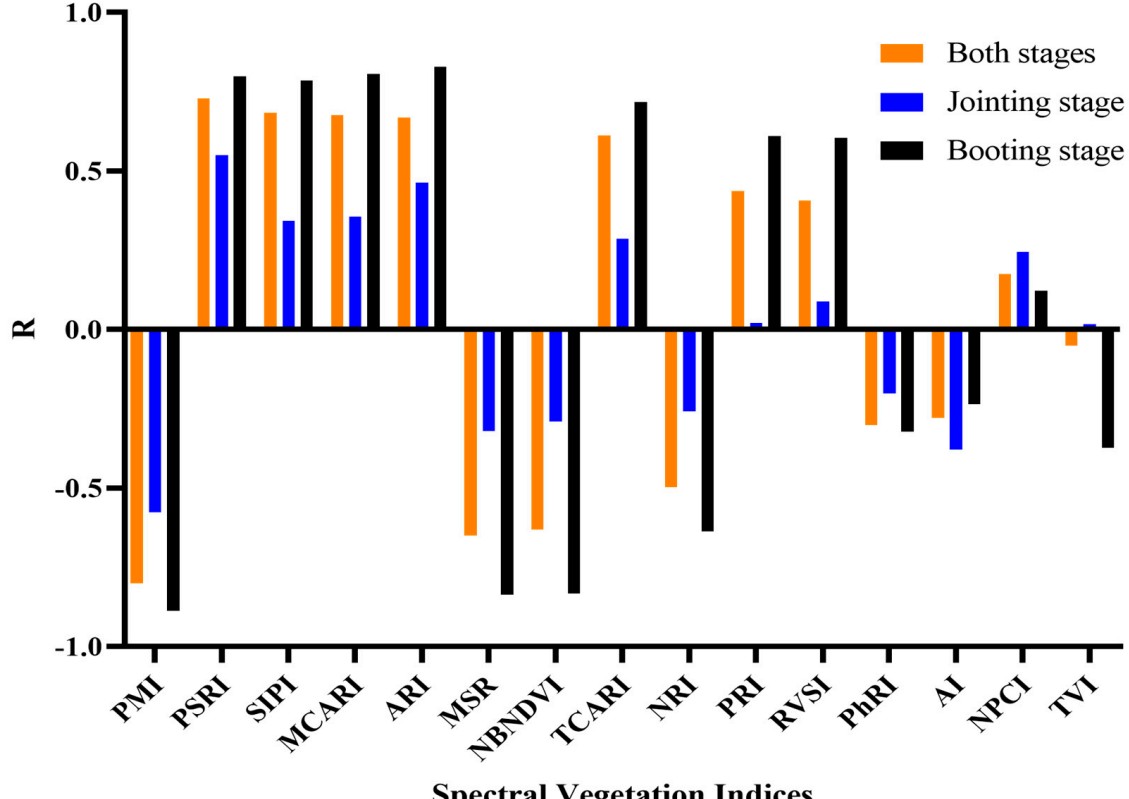

**Figure 5.** Response of spectral vegetation indices to DS at both and single growth stages.

### 3.2.3. Extraction of Texture Features

Because of the redundant information in hyperspectral images, here texture features of images at the six selected sensitive wavelengths were extracted, for the further development of NDTIs. Eight texture features for each spectral band and in total 48 texture features were adopted. Figure 6 shows one of the texture images at the 533.5 nm wavelength. Texture images at same wavelength are various due to different texture features.

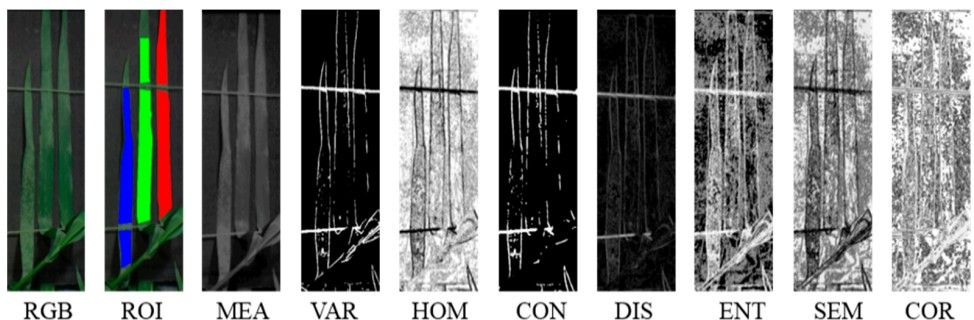

**Figure 6.** Texture images at wavelength of 533.5 nm. Note: MEA = mean; VAR = variance; HOM = homogeneity; CON = contrast; DIS = dissimilarity; ENT = entropy; SEM = second moment; COR = correlation).

The Pearson correlation coefficient r was used to examine the relationships between the texture features and the observed DS at different growth stages (Figure 7). The results showed that the texture features were poorly correlated with DS at the jointing stage but highly correlated at the booting stage. Whether it was at the jointing or the booting stage, 20 textures, namely homogeneity (HOM), dissimilarity (DIS), entropy (ENT), and second moment (SEM) at the wavelengths of 523.6, 528.5, 533.5, 659.2, and 661.7 nm were significantly correlated with DS.

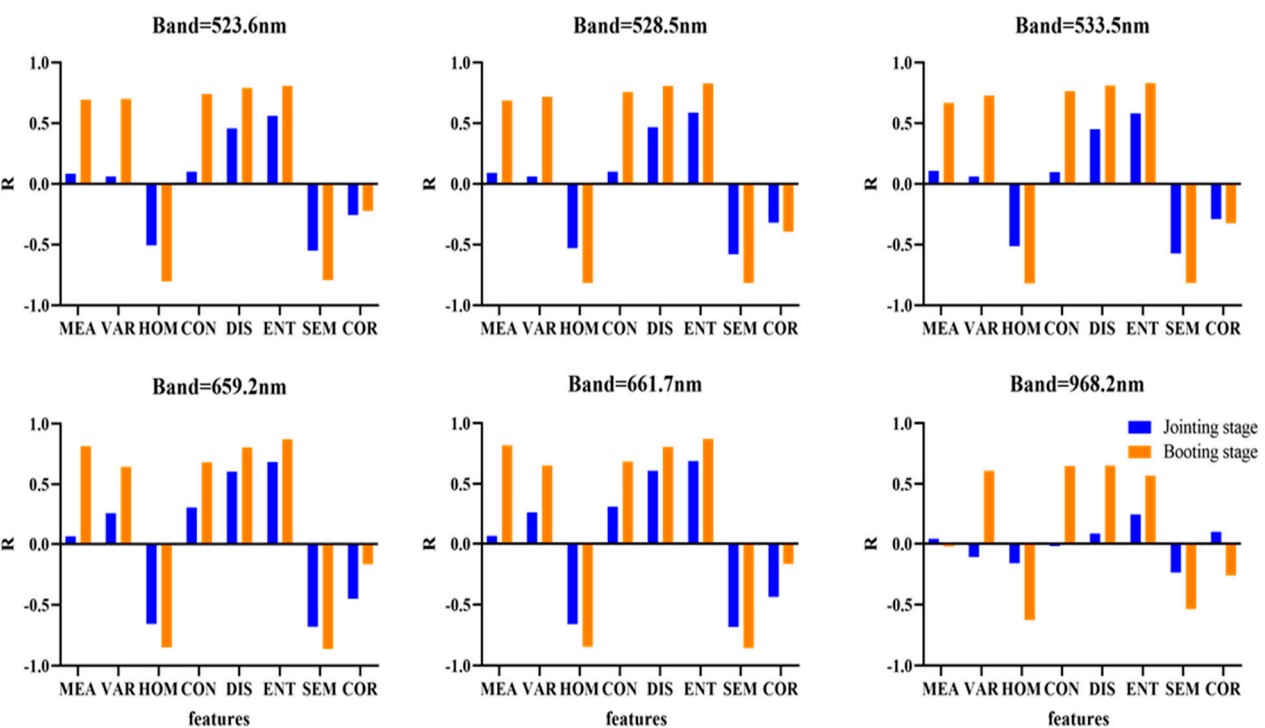

**Figure 7.** Pearson correlation analysis between texture features and DS.

### 3.2.4. Calculation of Normalized Difference Texture Indices (NDTIs)

NDTIs were constructed to improve the performance of texture analysis in disease recognition as the texture features did not exhibit consistent relationships to DS at two growth stages. Based on the aforementioned 48 texture features, 2256 NDTIs were realized by combining all two possible texture features, and the significance of those NDTIs were analyzed by correlation analysis. Among the 2256 NDTIs, only 10 NDTIs were sensitive to DS at different growth stages and throughout both growth stages, as listed in Figure 8. Ten best performing NDTIs were constructed by a combination of the following seven texture features, namely the second moment (SEM) at 659.2 nm and 661.7 nm and the homogeneity (HOM) at 523.6 nm, 528.5 nm, 533.5 nm, 659.2 nm, and 661.7 nm. Compared with texture features in Figure 7, the NDTIs are better correlated to disease severity. Thus, these NDTIs were selected for further construction of the disease detection model.

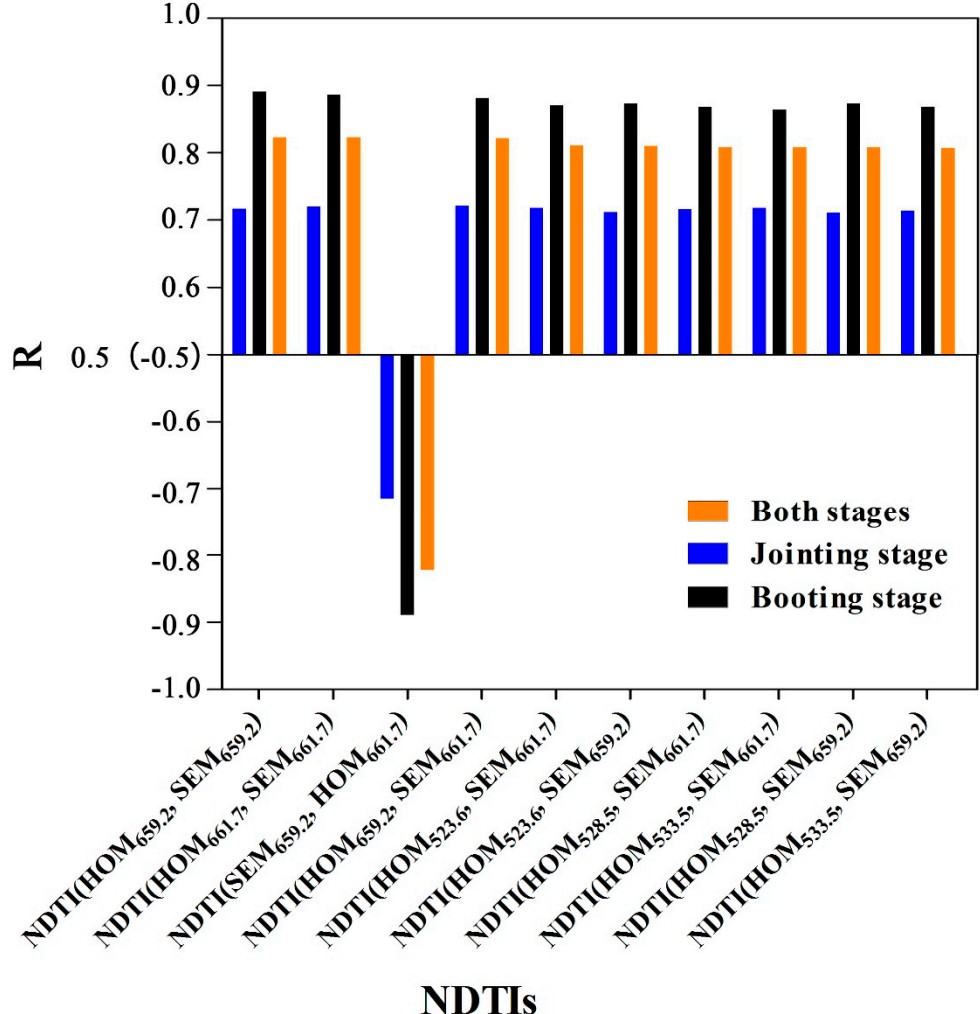

**Figure 8.** The correlation coefficient of the top 10 normalized difference texture indices (NDTI) which are highly correlated with DS.

### 3.3. PLS-LDA Model for Classifying the Healthy and Diseased Leaves

3.3.1. Evaluation of PLS-LDA Model Based on Different Selected Sensitive Features

The PLS-LDA model was built to evaluate the performance of different indices in terms of detecting wheat PM at different growth stages. The classification results were evaluated both for calibration and validation accuracies, as listed in Table 3. Regardless of growth stage, spectral vegetation indices achieved better classification accuracies than texture indices. However, the model based on the combination of VIs and NDTIs performed best in

comparison with that of VIs and NDTIs when used alone as input features. The combination of VIs and NDTIs significantly increased the classification accuracy, yielding calibration accuracies of 75.34%, 80.72%, and 76.46% and validation accuracies of 74.88%, 78.93%, and 76.23% at jointing stage, booting stage, and throughout both stages, respectively. The classification accuracies at booting stage were superior, followed by that at both stages and jointing stage, respectively.

**Table 3.** Classification results of PLS-LDA model based on different features.

| Dataset | Inputed Features | Features | Calibration Accuracy (%) | | | Validation Accuracy (%) | | |
|---|---|---|---|---|---|---|---|---|
| | | | Healthy | Infected | Overall | Healthy | Infected | Overall |
| Both stages | VIs | 6 | 76.49 | 78.26 | 77.13 | 74.19 | 74.00 | 74.22 |
| | NDTIs | 10 | 72.63 | 67.70 | 70.85 | 72.53 | 65.96 | 70.18 |
| | VIs & NDTIs | 16 | 77.17 | 75.16 | 76.46 | 77.62 | 73.72 | 76.23 |
| Jointing stage | VIs | 6 | 71.71 | 76.06 | 73.09 | 72.70 | 69.64 | 72.65 |
| | NDTIs | 10 | 73.68 | 73.24 | 73.54 | 73.47 | 67.80 | 72.23 |
| | VIs & NDTIs | 16 | 76.97 | 71.83 | 75.34 | 74.63 | 74.43 | 74.88 |
| Booting stage | VIs | 6 | 85.71 | 62.22 | 76.23 | 82.36 | 63.29 | 75.36 |
| | NDTIs | 10 | 75.19 | 63.33 | 70.40 | 77.42 | 63.36 | 70.30 |
| | VIs & NDTIs | 16 | 84.21 | 75.56 | 80.72 | 91.97 | 61.40 | 78.93 |

Note: VIs = vegetation indices; NDTIs = normalized difference texture indices; VIs & NDTIs = combination of VIs and NDTIs.

### 3.3.2. Classification of Healthy and Diseased Leaves at Early Stage after Inoculation

As demonstrated in Section 3.3.1 (Table 3), the model based on combination of VIs and NDTIs revealed better results. So, further for classification of healthy and diseased leaves as early as possible after inoculation, we established the PLS-LDA model based on combined features (VIs & NDTIs), on datasets of different days after inoculation (DAI) of each growth stage (Table 4). At the jointing stage, healthy and diseased leaves could be better distinguished on 6 DAI with an average DS of 3.9%, as the overall classification accuracy on this day approached more than 85% with a kappa value of 0.73 and the accuracies for healthy and diseased leaves were 82.35% and 100%, respectively. While, at booting stage, on 3 DAI the overall accuracy reached more than 85% with a kappa value of 0.73 and the accuracies for healthy and diseased leaves were recorded as 81.25% and 100%, respectively. The mean disease severity on this day was noted as 1.1%. Therefore, it can be concluded that wheat PM can be better diagnosed at an early-stage with a maximum of up to 3–6 DAI or at a mean DS of around 1–9%, by using combined (VIs & NDTIs) sensitive features as an input in classification model.

**Table 4.** PLS-LDA classification results of healthy and diseased leaves based on combined spectral and texture indices (VIs & NDTIs) at the jointing and booting stage.

| | | Classification Accuracies (%) | | | | | | | |
|---|---|---|---|---|---|---|---|---|---|
| Growth stage | DAI | 4 | 5 | 6 | 7 | 8 | 10 | 11 | 12 |
| | T (°C) | 21.5 | 21 | 17.5 | 11 | 13 | 12 | 8.8 | 15.8 |
| | DS/State | 1% | 1.3% | 3.9% | 8.1% | 16.6% | 23.1% | 24.9% | 25% |
| Jointing stage | Healthy | 56.25 | 57.14 | 82.35. | 87.5 | 84.62 | 87.5 | 100 | 100 |
| | Diseased | 100 | 100 | 100 | 87.5 | 100 | 100 | 90.91 | 90.91 |
| | OA (%) | 63.16 | 64.71 | 87.50 | 87.50 | 90.91 | 91.67 | 95.65 | 95.65 |
| | Kappa | 0.29 | 0.32 | 0.73 | 0.73 | 0.82 | 0.82 | 0.91 | 0.91 |
| Growth stage | DAI | 3 | 4 | 5 | 7 | 9 | 10 | 11 | 12 |
| | T (°C) | 27.7 | 25.8 | 31 | 31.5 | 30.8 | 32.4 | 29.3 | 33.2 |
| | DS/State | 1.1% | 6.2% | 9.6% | 13.5% | 25.6% | 25.7% | 31.9% | 45.4% |
| Booting stage | Healthy | 81.25 | 88.89 | 92.31 | 100 | 91.67 | 100 | 100 | 100 |
| | Diseased | 100 | 100 | 100 | 100 | 100 | 100 | 100 | 100 |
| | OA | 86.96 | 94.44 | 95.83 | 1 | 95.45 | 1 | 1 | 1 |
| | Kappa | 0.73 | 0.89 | 0.92 | 1 | 0.91 | 1 | 1 | 1 |

Note: DAI indicates days after inoculation, T is mean temperature of green house in degree centigrade, DS is mean disease severity, Healthy and Diseased indicate classification accuracies of inoculated and non-inoculated wheat leaves, OA is overall classification accuracy.

### 3.4. PLSR Model for Estimating the Disease Severity

For estimating the disease severity, the aforementioned six sensitive vegetation indices, 10 texture indices and 16 combined features (i.e., VIs & NDTIs) were used to construct the partial least square regression (PLSR) model with 10-fold cross-validation (Table 5). Across growth stages, the $R^2$ and *RMSE* results of booting stage were superior regardless of input features, followed by pooled data of both stages and then jointing stage, respectively. Based on input features, the model constructed by using VIs was comparatively better in the validation model than that of the model constructed by using NDTIs. However, the model constructed by using combined features (i.e., VIs & NDTIs) yielded significantly better results, both for calibration and validation, than the models separately based on VIs or NDTIs.

**Table 5.** Results of partial least-squares regression (PLSR) models based on different inputted features.

| Growth Stage | Inputted Features | Number of Features | Calibration | | Validation | | |
|---|---|---|---|---|---|---|---|
| | | | $R^2$ | *RMSE* | $R^2$ | *RMSE* | *RRMSE* |
| Both stages | VIs | 6 | 0.687 | 14.166 | 0.660 | 14.761 | 0.597 |
| | NDTIs | 10 | 0.694 | 14.001 | 0.649 | 14.992 | 0.606 |
| | VIs & NDTIs | 16 | 0.748 | 12.711 | 0.722 | 13.356 | 0.540 |
| Jointing stage | VIs | 6 | 0.527 | 15.166 | 0.431 | 16.636 | 0.924 |
| | NDTIs | 10 | 0.531 | 15.102 | 0.344 | 17.872 | 0.993 |
| | VIs & NDTIs | 16 | 0.619 | 13.624 | 0.532 | 15.162 | 0.842 |
| Booting stage | VIs | 6 | 0.831 | 9.990 | 0.792 | 11.511 | 0.384 |
| | NDTIs | 10 | 0.815 | 11.374 | 0.747 | 13.230 | 0.443 |
| | VIs & NDTIs | 16 | 0.855 | 10.060 | 0.818 | 11.320 | 0.377 |

Figure 9 shows DS estimation models based on combined features (i.e., VIs & NDTIs), by using the validation datasets of different growth stages. It can be observed that the measured DS is close to the predicted DS which lies along line of concordance (1:1 solid line), confirming the accuracy of the DS estimation model. The performance of the estimation model at booting dataset was superior, i.e., the $R^2$ values between measured and estimated DS was 0.818 and *RMSE* was 11.320, followed by pooled data of both stages ($R^2$ = 0.722, *RMSE* = 13.356). While the model performance at jointing stage was relatively poor ($R^2$ = 0.532, *RMSE* = 15.32) and the predicted DS was overestimated. The above results suggest that the PLSR model based on the combined features of VIs and NDTIs performed better for detecting wheat PM disease severity.

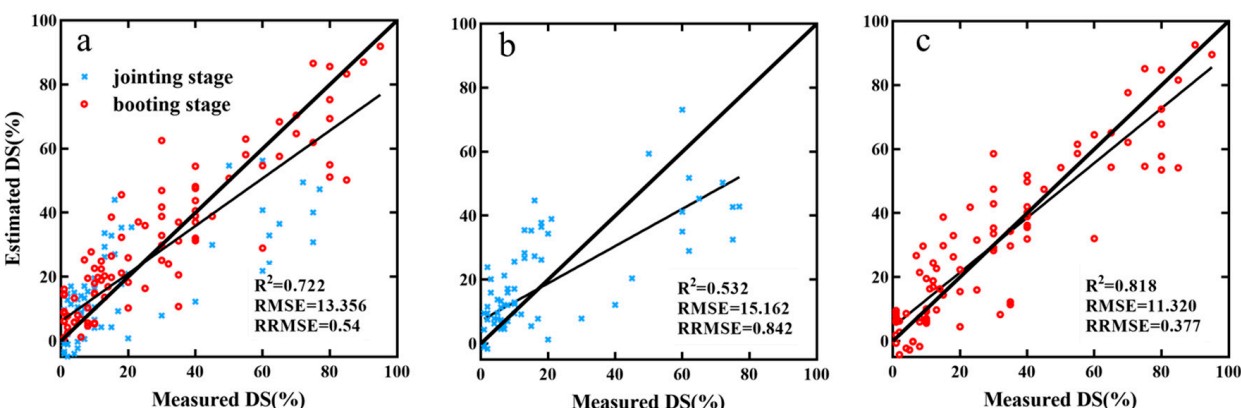

**Figure 9.** Comparison of measured and estimated wheat powdery mildew DS based on combined features of VIs & NDTIs, at both stages (**a**), jointing stage (**b**), and booting stage (**c**).

## 4. Discussion

### 4.1. Why the Selected Features Are Rational?

In this study, six sensitive spectral bands (533.5, 659.2, 528.5, 661.7, 968.2, and 523.6 nm) were selected by SPA method to reduce the hyperspectral dimensions, in agreement with previous studies revealing sensitive spectral regions to wheat PM [30]. Among these spectral bands, three wavelengths at around 550 nm are sensitive to photosynthetic pigments and have great potential for early disease monitoring, in accordance with the findings of Maimaitiyiming [47]. When a pathogen infects wheat, the physiological parameters respond accordingly, such as decreased chlorophyll content, which affects the spectral reflectance in the visible light region. As shown in Figure 10, the chlorophyll content of healthy and diseased leaves on different days after infection were analyzed by analysis of variance ($p < 0.05$). At the jointing stage, the chlorophyll content of healthy and diseased leaves differed significantly on day 8 after infection. At the booting stage, the chlorophyll content of healthy and diseased leaves differed significantly on day 6 after infection. Decreased chlorophyll content weakened the ability to absorb light, therefore resulting in higher reflectance in the VIS range.

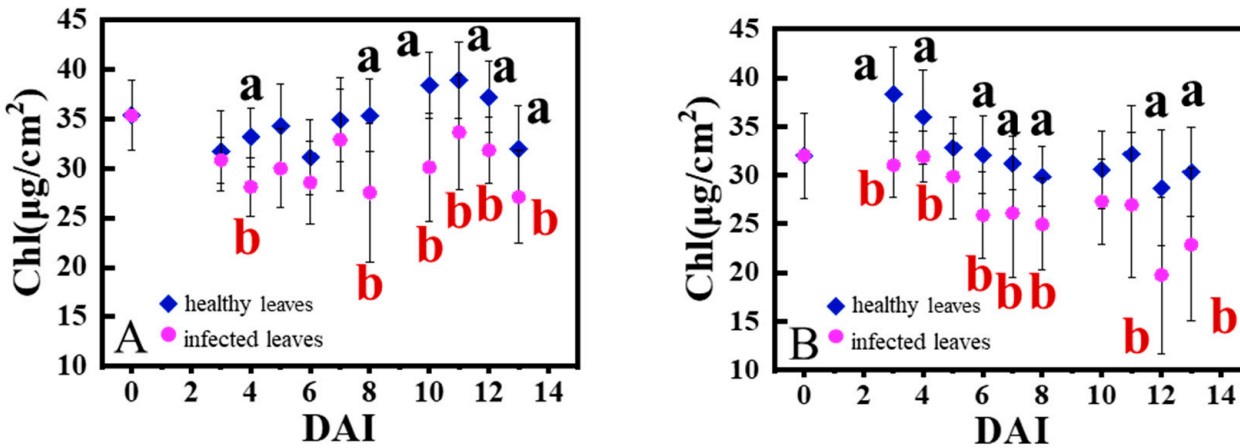

**Figure 10.** Changes and comparison of chlorophyll content between healthy leaves and diseased leaves infected at jointing stage (**A**) and booting stage (**B**) on the day after inoculation (DAI). Note: ab represents a difference that is significant ($p < 0.05$).

In the near-infrared range, the reflectance is affected by leaf anatomy, which corresponds to the sensitive wavelength at 968.2nm in this study. As shown in Figure 11, the chloroplast structure (marked by "Ch") in leaves with different DS on 13 DAI was observed using a transmission electron microscope. As the DS increased, the chloroplast in the wheat leaves gradually swelled in the cells. Cavities of varying sizes appeared surrounded by a single layer of membrane, and the chloroplast disintegrated eventually. The cells in the normal leaves form multiple scattering of light resulting in high reflectance, while powdery mildew affects the internal structure of leaves, thus leading to lower reflectance of infected leaves.

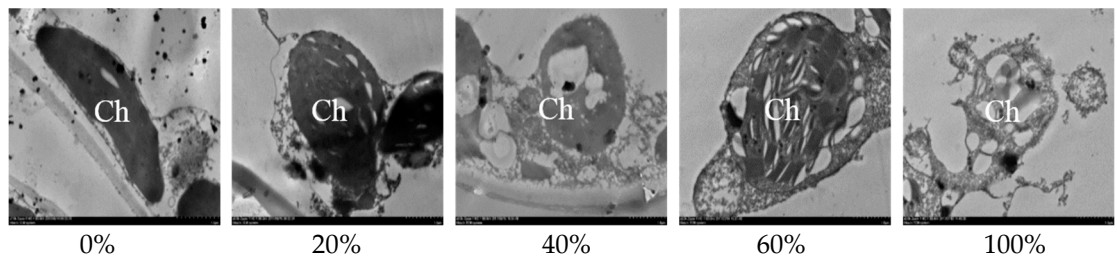

**Figure 11.** Chloroplast structure of wheat at different disease severity (DS) after infection.

For texture features, Xie et al. [48] first compared spectral and texture features for detecting different diseases on tomato leaves, and the results obtained by texture features (60.2%) were slightly worse than those obtained by reflectance (97.1%) in classification accuracy. Al-Saddik et al. [23] used hyperspectral images to detect yellowness and esca in grapevines by combining spectral and texture data with an overall accuracy of 99% for both of the diseases. The adoption of NDTI in disease detection of this study is inspired by Zheng [33], who combined NDTIs and VIs to estimate above ground crop biomass with $R^2$ of 0.78. However, the features selected in this study are specific to the wheat disease PM. When this methodology is used to monitor diseases on other crops, it would need to re-select the sensitive spectral features and corresponding texture features, as the symptoms (shape and color) of each disease are different.

### 4.2. What Is the Reason of Detection Performance at Varied Growth Stages?

The detection at the booting stage performed significantly better than that of jointing stage, which might be due to the different rates of disease propagation at two growth stages. Most of the DS values for diseased leaves obtained after infection at the jointing and booting stages were 1–30% and 1–50%, respectively. The factors which affect the development of wheat PM are environment conditions [49], variety resistance [3], and the number of pathogen spores [50]. The varieties, cultivation practices and the number of spores for inoculations used in the two growth stages were the same. Hence, the difference was mainly due to the different climatic conditions during the two inoculation periods. The inoculation interval between the two growth stages differed by 14 days. The inoculation time at the jointing stage was early spring, at this stage the temperature was still low and gradually rising, therefore the development of the disease was slow. At low temperature, the PM fungus tends to reproduce in small quantities and the physiological and biochemical characteristics of diseased plants may not be obvious. However, the disease developed rapidly at the booting stage because of the suitable environment conditions. It was the time of late spring and conditions of high temperature and humidity were optimal for disease development, consistent with the observations of Yao et al. [51]. Such disease favorable environmental conditions enhanced the rate of disease development at the booting stage, thus inducing obvious changes in the biophysical and biochemical parameters of plants which resulted in changed spectral responses [52]. Moreover, high lesion ratios on the leaf surfaces at booting stage changed the texture patterns of wheat leaves more severely thus, ensuring robust and accurate disease detection at booting stage.

### 4.3. How Early to Detect the Disease When the Combined Feature Is Applied?

This paper focuses on the methodology with regard to the feasibility of detecting the disease at its early stage or even before symptoms appear, at which it is difficult to acquire the weak signal due to the limited bands and course resolution. To the best of our knowledge, the present work is the first attempt to detect wheat PM by combining spectral vegetation indices and texture indices. Wheat PM lesions induce modification of color and surface properties of wheat leaves, and the texture information describes the intensity of the pixel's changes in an image. This phenomenon explains the robustness of the texture approach combined with the spectral approach in the better diagnosis of disease. Previous studies found that the spectral approach performed better when the DS is usually greater than 20%, with classification accuracy of about 87%. Meanwhile, the texture approach performed better in distinguishing DS of less than 20% with classification accuracy of 84%, but the accuracy by using a combination of texture and spectral features can reach up to 90% [53]. Therefore, combining spectral and texture information allows significantly better classification accuracy of diseased and healthy leaves and DS estimation at the early stage of wheat PM.

The early detection of disease on a daily basis after inoculation at different growth stages was carried out by combining the spectral and texture indices, and the classification accuracies of healthy and diseased leaves increased with increasing DS. On 6 DAI at jointing

stage, the classification of healthy and diseased leaves was robust and stable, when the average DS was 3.9%. While at the booting stage, disease was recognized well on 3 DAI with an average DS of 1.1%. As the DS increased, the spectral and texture features changed, and the classification accuracies improved as the days passed after inoculation. However, at 9 DAI of booting stage, a slight decrease in classification accuracy was observed. These slight variations in classification accuracies were due to the unbalanced number of healthy and diseased leaves. At the booting stage on 9th DAI, most of the leaves were infected, so the model tended to predict the diseased one, resulting in decreased accuracy of healthy leaves. In another aspect, the variations of chlorophyll contents in leaves are also consistent with the model's performance for disease detection, as the model correctly identified disease on 6 and 3 DAI (Table 4) with an OA of over 85% and chlorophyll contents significantly changed between healthy and diseased leaves on 8 and 6 DAI (Figure 10), at jointing and booting stages, respectively. Therefore, our results well support the idea that HI is an efficient and reliable method for plant disease detection at early stage. It is worth mentioning that ground-based hyperspectral analysis is not suitable for a large area, and it is too expensive to be used by average farmers compared with UAV platforms. Multi-spectral UAV platforms can monitor disease in large areas with stable and high efficiency but perform poorly at the early disease infection stage. In the future, we believe there would be more hyperspectral equipment mounted on UAV systems, however currently there is only one kind of UHD185 in the community.

## 5. Conclusions

In this paper, spectral and texture analysis of hyperspectral imaging technique were used to discriminate wheat powdery mildew without obvious symptoms and estimate corresponding disease severity. This study indicates that the combination of spectral and texture approaches is the sensitive feature for disease detection when building a PLS-LDA classification model and PLSR DS estimation model, which offered significantly improved accuracies in detection and quantification of wheat PM disease. The model based on selected sensitive features identified the disease even before the initiation of significant variations in the physiological and biochemical parameters of leaves after disease. Therefore, it can provide the basis for early detection, preventing and controlling the plant disease worldwide. The stable performance of the spectral and texture indices enabled the early detection of disease with more than 85% overall classification accuracies of PLS-LDA model, especially at early infection stage of 3–6 DAI with a DS of 1–6%. The DS can be well estimated by PLSR model with the $R^2$ value of 0.818 at booting growth stage. However, the present study is conducted on leaf scale with relatively few trials, and only used limited number of diseased and healthy leaves for classification on a daily basis after inoculation. In the future, we would evaluate the feature and model with a greater number of trials and data, and hope to expand to field and canopy level.

**Author Contributions:** Conceptualization, X.Y., H.L. (Hongyan Liu); methodology, H.L. (Haiyan Liu), H.L. (Hongyan Liu) and I.H.K.; software, H.L. (Hongyan Liu); validation, H.L. (Haiyan Liu); formal analysis, I.H.K.; investigation, I.H.K. and H.L. (Hongyan Liu); resources, W.L., X.W.; writing—original draft preparation, H.L. (Hongyan Liu) and H.L. (Haiyan Liu); writing—review and editing, H.L. (Haiyan Liu), X.Y., A.C., W.L., T.C., Y.T., Y.Z., W.C., supervision, X.Y.; project administration, X.Y.; funding acquisition, X.Y. All authors have read and agreed to the published version of the manuscript.

**Funding:** This work was supported by grants from the National Key Research and Development Plan of China (2019YFE011721), National Natural Science Foundation of China (31971780), the Key Projects (Advanced Technology) of Jiangsu Province (BE 2019383), 333 Project of Jiangsu Province (JS333), Collaborative Innovation Center for Modern Crop Production co-sponsored by Province and Ministry(CIC-MCP), the Priority Academic Program Development of Jiangsu Higher Education Institutions (PAPD), and the 111 project (B16026).

**Institutional Review Board Statement:** Not applicable.

**Informed Consent Statement:** Not applicable.

**Data Availability Statement:** Not applicable.

**Acknowledgments:** The authors would like to thank the reviewers for recommendations which improved the manuscript.

**Conflicts of Interest:** The authors declare no conflict of interest.

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
