# Peer review of "Early Detection of Powdery Mildew Disease and Accurate Quantification of Its Severity Using Hyperspectral Images in Wheat"

_remotesensing, doi:10.3390/rs13183612_

Round 1

Reviewer 1 Report

This version of the manuscript contains the corrections and additions that I requested earlier (first submitted version). However, some minor adjustments still need to be made. Because of this, my recommendation about the manuscript is Minor Revision. 

1) The authors should check the figures and tables again such as table 3, table 5 and figure 8. 

2) The authors should re-create Figure 5 and Figure 7. It is not easy to read the R values that were written on the columns. 

3) Please check page 16 line 511. I think it should be multispectral, not multiple spectral.  

4) Please check the whole manuscript there are some editing problems. 

Author Response

Response to Reviewer 1

September 6, 2021

Ms. Milica Kovacevic

Section Managing Editor                                

Remote Sensing

Dear Editor,

Thank you very much for your valuable comments regarding the revision on our manuscript titled as “Early Detect the Powder Mildew Disease and Accurately Quantify the Severity Using Hyperspectral Images in Wheat” (remotesensing-1359404). We have seriously and carefully revised the article according to the instructive comments from the reviewers and the editor, which was marked up using the “Tracking Changes” in the revised manuscript. Our detailed responses to the comments are provided on the attached files. We seriously considered and carefully revised all the issues, and all changes marked with red.

Sincerely yours,

Xia Yao,

Ph.D., Professor

Point 1: The authors should check the figures and tables again such as table 3, table 5 and figure 8.

Response 1: we have checked the figures and tables, and recreated Figure 8.

Point 2: The authors should re-create Figure 5 and Figure 7. It is not easy to read the R values that were written on the columns.

Response 2: Figure 5 and Figure 7 have been revised.

Point 3: Please check page 16 line 511. I think it should be multispectral, not multiple spectral. 

Response 3: It has been revised.

Point 4: Please check the whole manuscript there are some editing problems.

Response 4:  we have check the whole manuscript including the language and editing problems.

Reviewer 2 Report

Every comment has been properly answered.

Author Response

We have checked the whole manuscript including the the conclusions.

This manuscript is a resubmission of an earlier submission. The following is a list of the peer review reports and author responses from that submission.

Round 1

Reviewer 1 Report

This study presented a method to identify and evaluate powdery mildew of wheat using hyperspectral images. The manuscript is well-organized and written. However, the title is misleading and need to strengthen the justification as well.

Specific comments:

Title: Title is too general. Authors only focused only one disease not all diseases of wheat not this is a review article. Suggest using a title specific to this research.

Line 17: What you meant by 'model' is unclear.

21: Missing space between '&' and 'NDTIs'

102-103: Not clear. Rephrase.

117-118: By this definition and figure 1, looks like a RGB camera should suffice to estimate DS with some image processing. RGB camera is cheaper and easier to process than Hyperspectral camera. Have you tried to compare RGB and Hyperspectral images to estimate DS? Is there any accuracy increase? If so, what could have affected that? If there is no advantage of using hyperspectral camera for DS over RGB camera, how do you justify the use of Hyperspectral camera?

Fig 2: Remove the red underline from the word 'Hyperspectral'

208-209: Are there any other PLS methods?

Reviewer 2 Report

In this manuscript, spectral and textural analysis of lab collected hyperspectral data were used to discriminate wheat powderry mildew without obvious symptoms and estimate disease severity. My recommendation about the manuscript is reconsider after major revision. 

1) In introduction section, the authors should provide more references especially related with SPA. Furthermore,  the authors should provide detailed explanation about the difference of the study than the published articles.  

2) The authors should provide photos of the experimental pots, especially in different stages. 

3) The authors should provide information about the selected metrics such as R2, RMSE and RRMSE. Why these metrics were selected for the study? 

4) There are many tables in the manuscript. It is not easy to follow all the results from these tables. The authors should provide some graphs to clearly share the results. 

5) In page 10, line 290, Fig. 4 must be Fig.5. 

6) For Fig 6: The authors should provide the estimation models for all growth stages, inputted features and number of the features with 1:1 line (in different color) and all metrics (R2, RMSE and RRMSE). 

Reviewer 3 Report

This study presents an interesting hyperspectral imaging technique based on spectral and texture analysis in order to detect powdery mildew in wheat.  In general the research idea is good and I strongly support this manuscript, however I have some concerns.

  • The literature review is scarce, I think that would be good to mention some other studies even in other remote sensing techniques or in other crops i eg:

RGB based analysis:

Neumann, M., Hallau, L., Klatt, B., Kersting, K., and Bauckhage, C. 2014. Erosion band features for cell phone image based plant disease classification. Pages 3315-3320 in: Proceeding of the 22nd International Conference on Pattern Recognition

Bock, C. H., Poole, G. H., Parker, P. E., and Gottwald, T. R. 2010. Plant disease severity estimated visually, by digital photography and image analysis, and by hyperspectral imaging. Crit. Rev. Plant Sci. 29:59-107

Behmann, J., Mahlein, A.-K., Rumpf, T., Römer, C., and Plümer, L. 2014. A review of advanced machine learning methods for the detection of biotic stress in precision crop protection. Precis. Agric. 16:239-260.

Multiespectral analysis:

Zhang, D., Zhou, X., Zhang, J., Lan, Y., Xu, C., & Liang, D. (2018). Detection of rice sheath blight using an unmanned aerial system with high-resolution color and multispectral imaging. PloS one, 13(5), e0187470.

Viera-Torres, M., Sinde-González, I., Gil-Docampo, M., Bravo-Yandún, V., & Toulkeridis, T. (2020). Generating the baseline in the early detection of bud rot and red ring disease in oil palms by geospatial technologies. Remote Sensing12(19), 3229.

Mirik, M., Jones, D. C., Price, J. A., Workneh, F., Ansley, R. J., & Rush, C. M. (2011). Satellite remote sensing of wheat infected by wheat streak mosaic virus. Plant disease95(1), 4-12.

Hyperspectral analysis:

Mahlein, A. K., Alisaac, E., Al Masri, A., Behmann, J., Dehne, H. W., & Oerke, E. C. (2019). Comparison and combination of thermal, fluorescence, and hyperspectral imaging for monitoring fusarium head blight of wheat on spikelet scale. Sensors, 19(10), 2281.

Adam, E., Deng, H., Odindi, J., Abdel-Rahman, E. M., & Mutanga, O. (2017). Detecting the early stage of phaeosphaeria leaf spot infestations in maize crop using in situ hyperspectral data and guided regularized random forest algorithm. Journal of Spectroscopy, 2017.

Bravo, C., Moshou, D., West, J., McCartney, A., and Ramon, H. 2003. Early disease detection in wheat fields using spectral reflectance. Biosystems Eng. 84:137-145

Between others…

I think It would be good to mention the adventages and disadventages between RGB, multi and hyperspectral imagery for plant diseases analysis.

  • About methodology, I am concerned about the performance in plant desease detection, for example compared with an UAV imagery. Do you think that this multispectral technique could be relevant for the average farmer? I consider that must be explained elsewhere.
  • The readability of the equations and the figures must be improved.
  • Table format can be improved, in example, tables 3, 4, 5, 6, 7 take too much space.
  • In discussion I would like to see if this study can be applied in other crops, with other diseases.
  • Could be feasible to use this hyperspectral sensor onboard of an UAV, in order to enlarge the study area?
  • Are you sure that between 6 and 3 days are soon enough to avoid the crop performance decline?
  • In discussion must be more similar works using textures and Vis, to compare with yours.
  • Figure 7 is confusing and annoying, I suggest to improve it in order to avoid clarification notes.
  • In figure 7, for jointing stage, in the 4th DAI, it detects a significant difference between healthy and diseased leaves but in the 5th 6th and 7th day It doesn`t ¿Why?
  • In that sense, in order to apply the ANOVA, data must perform, normality, homoscedasticity and independence ¿Does the data perform than? It must be mentioned.
  • The same for Pearson Correlation Coefficient R2
  • I suppose that for texture features calculation must be a mathematic equation or at list other authors that express this, please add these equation or cite the original font.